# Exciton dissociation in organic solar cells:
# An embedded charge transfer state model

Jouda Jemaa Khabthani[1][*], Khouloud Chika[1], Alexandre Perrin[2] and Didier Mayou[2]

**1** Université Tunis El Manar, Faculté des Sciences de Tunis,
Laboratoire de Physique de la Matière Condensée, 1060 Tunis, Tunisia
**2** Université Grenoble Alpes, CNRS, Institut NEEL, F-38042 Grenoble, France

[*] jouda.khabthani@fst.utm.tn

## Abstract

Organic solar cells are a promising avenue for renewable energy, and our study introduces a comprehensive model to investigate exciton dissociation processes at the donor-acceptor interface. Examining quantum efficiency and emitted phonons in the charge transfer state (CTS), we explore scenarios like variations of the environment beyond the CTS and repulsive/attractive potentials. The donor-acceptor interface significantly influences the injection process, with minimal impact from the environment beyond the CTS. Attractive potentials can create localized electron states at the interface, below the acceptor band, without necessarily hampering a good injection at higher energies. Exploring different recombination processes, including acceptor-side and donor-side recombination, presents distinct phases for the injection process versus the initial energy of the electron and the recombination rate. Our study highlights the important role of the type of recombination in determining the quantum efficiency and the existence of hot or cold charge transfer states. Finally, depending on the initial energy of the electron on the donor side, three distinct injection regimes are exhibited. The present model should be helpful for optimizing organic photovoltaic cell interfaces, highlighting the critical parameter interplay for enhanced performance.



# 1 Introduction

High-performance excitonic solar cells such as heterojunction bulk organic solar cells (OSCs) demand efficient charge carrier excitation, separation, and extraction. Specifically, the transfer of charge at the interface between different electron or hole-transporting materials is a crucial step in these photovoltaic devices [1–7]. In the case of organic solar cells (OSCs), following the absorption of photons, excitons (bound electron-hole pairs) are created in a confined area on the donor side of the OSC. Therefore, the excitons must migrate to the interface between the donor and acceptor to separate into free electrons and holes and they need to overcome their mutual Coulomb attraction [8,9] which is larger than the thermal energy room.

A significant limiting factor toward the development of more efficient OSCs is the lack of a comprehensive understanding of their working principles. For instance, despite significant efforts on both the experimental and theoretical sides, the mechanism responsible for exciton dissociating into free charges is still poorly understood. Various phenomena have been recognized as aiding in charge separation (i.e., exciton dissociation), including built-in electric fields at donor-acceptor (D:A) interfaces, the delocalization of excitons and charge carriers, energetic variability, and structural disorder [7–13]. Beyond these crucial aspects related to electronic states at the interface and electrostatic effects, the charge transfer process is dynamic. Consequently, it is important to describe the dynamics of this process, such as the excitation of vibrational modes or the competition between charge separation and charge recombination between the electron and the hole [14].

On the theoretical side, various numerical methods have been proposed for a fully microscopic understanding of the charge separation mechanism, including Monte Carlo simulations [15,16], exact diagonalization [17], and time-dependent density functional theory [18–20]. Additionally, several models have been developed [11,21], such as the Braun-Onsager analytic model [22,23], the Marcus theory [24] and its related approaches, and ab initio electronic structure calculations [25–27]. However, none of these models have adequately clarified the processes occurring at the donor-acceptor interface during charge transfer. New complementary approaches are needed.

In this article, we propose a quantum model for charge separation at a donor-acceptor interface, focusing on electron-phonon coupling and the partially coherent nature of the electron. The central concept is the CTS, where the electron moves to the acceptor side and couples with local vibration modes. This coupling allows the electron to excite phonons, leading to a

vibrationally hot CTS if multiple phonons are excited, or a cold CTS if none are. The efficiency of charge separation is debated in relation to energy release on these modes [11–13, 28–33].A first analysis of charge injection was presented in our previous work [34]. Here we enlarge our approach by considering several types of recombination processes, various type of environments, attractive or repulsive potential and by analysing more deeply the process of injection and its interdependence with phonon emission on the CTS. In the context of the model our numerical treatment of the dynamics combines the quantum scattering theory and the dynamical Mean Field Theory (DMFT) and is exact. [17, 35–38].This allows for the treatment of the strong coupling regime between the electron and vibrational modes, in contrast to perturbative theories.

The paper is organized as follows. We begin with a presentation of the embedded CTS model and then discuss briefly the formalism that allows to compute the output of the dynamical injection process. The physics of the injection process is analyzed firstly for simple cases and then with the full complexity of the model. We then proceed to a global discussion of the results where we distinguish three regimes for injection, according to the value of the initial energy of the electron.

## 2 The embedded CTS model

### 2.1 Global scheme

Our charge separation model, depicted in Figure (1), provides a straightforward explanation of the injection process occurring at the donor/acceptor interface. It focuses on optical modes with frequencies higher than the thermal energy at room temperature, ensuring that all vibration modes are initially unoccupied when the charge injection process commences.

This model takes into account various injection processes between the donor and the acceptor. The electron on the donor site I can either recombine with the hole on site I with probability $\Phi_R^I$ or inject itself into the CTS site 0 in the acceptor with probability $\Phi_A^I$. From there, it can be injected into environmental channels or recombination channels. Alternatively, the electron can create a phonon and move to site 1 on the vertical axis, representing a state with one phonon of the vibration mode associated with the CTS. At each phonon mode level (vertically), there are three possibilities: Either the electron injects into the environment, recombines, or creates a phonon and moves up one level on the vertical axis. The quantities $\Phi_A^n$ and $\Phi_R^n$ represent the probability of injection into the environmental channel (with $n$ phonons emitted on the CTS) and the probability of injection into the recombination channel (with $n$ phonons emitted on the CTS), respectively.

### 2.2 Model for the environment on the acceptor side

The state $|I>$ corresponds to the excited singlet state, serving as the initial stage from which the electron can transition into the charge transfer state (CTS) through the hopping integral $m$. In state $|I>$, the electron resides in the LUMO orbital on the donor side, while the hole occupies the HOMO orbital. To model the environment of the acceptor, which constitutes the charge transfer site as well as the rest of the acceptor, we used the Bethe lattice geometry. The electron is capable of traversing between sites $i$ within successive layers $L = 0, 1, 2, ...$ (where layer $L = 0$ represents the CTS, see figure 2) and can induce phonon excitations across all sites within a given layer. We assume the sites $i$ to be distributed on a Bethe lattice, known for accurately reproducing the local environment of a compact system. For simplicity, we adopt the standard limit of infinite coordination for this Bethe lattice, wherein the mean-field solution provided by the DMFT is exact.

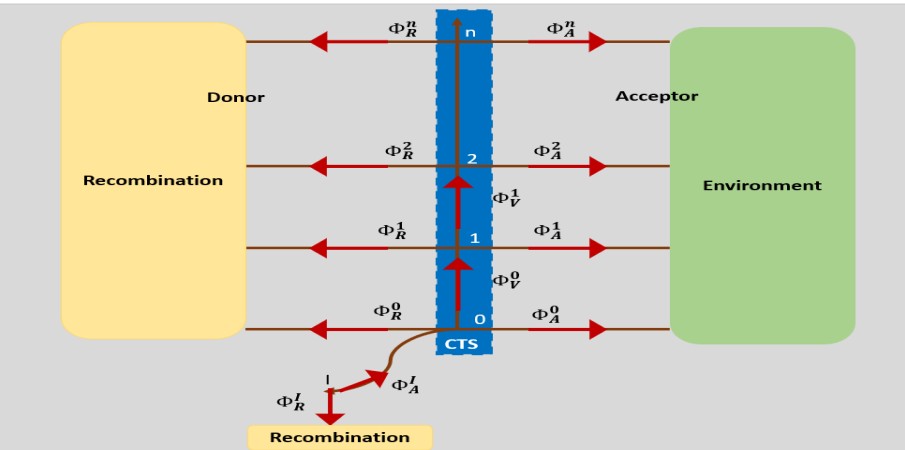

Figure 1: A simple illustration of the injection process at the donor/acceptor interface.

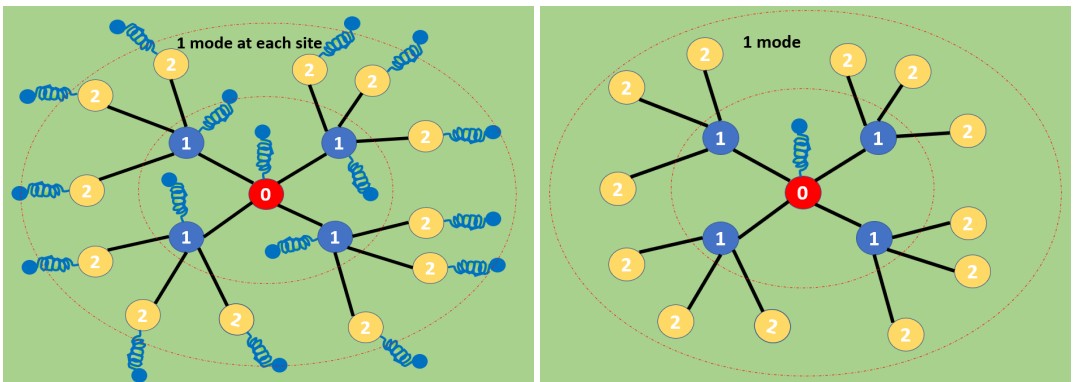

Figure 2: Two models for the environment on a Bethe lattice geometry with 4 nearest-neighbors. Model A (left) with a vibration mode at each site, and Model B (right) with a vibration mode only at the charge transfer site. The layers $L = 0, 1, 2$ are represented and $L = 0$ is the CT state.

To examine the effect of phonon modes, we employed two models: In the first model, a phonon mode is present on each site (Model A). In the second model, a phonon mode is present on the charge transfer site while the rest of the Bethe lattice is empty (Model B). In Figure 2, the two models are depicted.

## 2.3 Recombination and electrostatic potential

There are several electron-recombination processes with the hole on the donor site. The electron, when it is on the donor site in the excited state, can recombine with the hole in the ground state, also located on the donor site. Similarly, when it injects into the acceptor site, the electron can recombine with the hole it left behind. Figure 3 illustrates the two recombination processes: On the left, recombination from the donor site, and on the right, recombination from the acceptor site.

Several mechanisms can lead to the genimate recombination between the electron in the acceptor side and the hole left in the donor side. This can be for example photon emission or multiple emission of low energy phonons which lead to a transfer of the electron into localized states near the interface and then to recombination with the hole [14]. If recombination is accompanied by the emission of a photon, the continuum of states accessible by this recom-

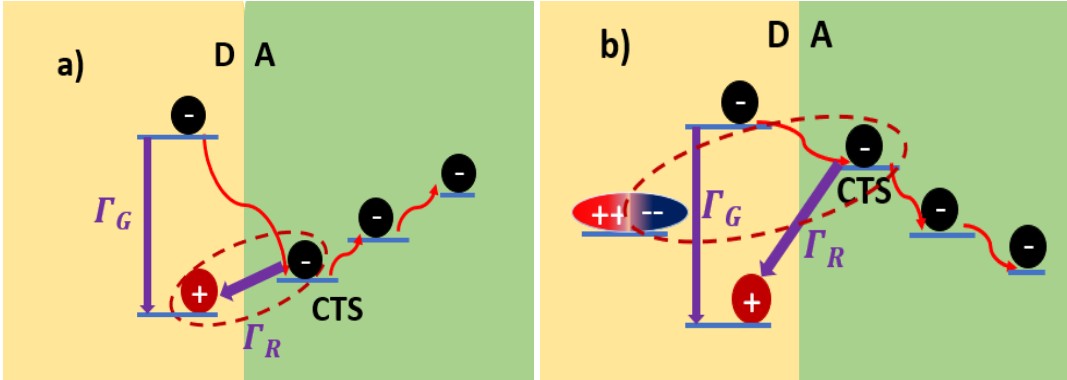

Figure 3: The two types of potentials are illustrated: a) the attractive potential, b) the repulsive potential. The two recombination processes: $\Gamma_R$: Recombination on the acceptor site, $\Gamma_G$: Recombination on the donor site.

bination is broad in energy compared to a typical polaronic bandwidth. However, for the other recombination processes the continuum of excitations may not necessarily be broad compared to a typical polaronic bandwidth. Thus, we distinguish between these two different types of injection, and we show that this can have importance on the course of injection.

There are two types of potential at the donor/acceptor interface: Either an attractive potential V<0 between the electron on the CTS site and the hole on the donor site, or a repulsive potential V>0 between the electron on the CTS and the negative charges of the molecules in the donor. In Figure 3, both types of potential are illustrated: On the left, the attractive potential, and on the right, the repulsive potential and the two recombination processes: $\Gamma_R$: Recombination on the acceptor site, $\Gamma_G$: Recombination on the donor site.

## 2.4 Holstein Hamiltonian

The Hamiltonian $H$ of our model takes into account a tight-binding hopping model for the electron and includes the essential coupling with local vibration modes, as is common for molecular solids [21]. $H$ is of a Holstein type [39, 40] and takes the form:

$$
\begin{aligned}
H = {}& \varepsilon_I c_I^+ c_I - m(c_I^+ c_0 + c_0^+ c_I) + \sum_i \frac{V}{L_i + 1} c_i^+ c_i \\
& + \sum_i \hbar \omega a_i^+ a_i - \sum_{i,j} J_{i,j} \left( c_i^+ c_j + c_j^+ c_i \right) \\
& + \sum_i g_i c_i^+ c_i \left( a_i^+ + a_i \right) + H_R,
\end{aligned}
\tag{1}
$$

where the index $i$ represents distinct sites, with $i = 0$ denoting the CTS. $\varepsilon_I$ stands for the energy of the incoming electron of a molecule located at the donor site. The hopping parameter $m$ between the donor site and the CTS is considered weak compared to the pure electronic bandwidth $4J$, allowing us to approach the limit $m \rightarrow 0$. For a given site $i$, the creation (annihilation) operators for electrons are denoted as $c_i^+$ ($c_i$). The electrostatic potential at site $i$, situated on layer $L_i$, is expressed as $\frac{V}{L_i+1}$, resulting from electron interaction with the hole or with interfacial charges and characterized by the parameter $V$. The creation (annihilation) operators for the local vibration mode at site $i$ are represented by $a_i^+$ ($a_i$), with $\hbar \omega$ representing the phonon energy ( we consider $\hbar = 1$). The hopping matrix elements $J_{i,j}$ between nearest neighbors $i$ and $j$ on the Bethe lattice are expressed from $J$. In our approach, we adopt the standard limit of infinite coordination $K$ for the Bethe lattice, while simultaneously imposing

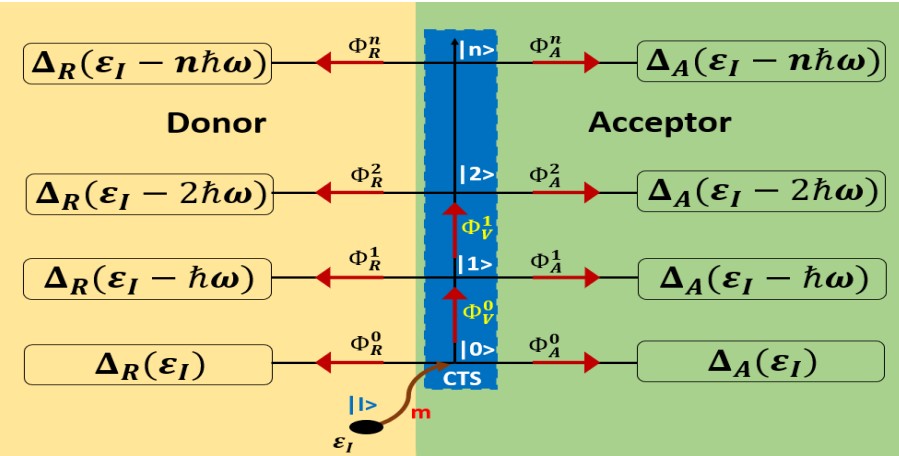

Figure 4: Schematic representation for the charge injection process in the Hilbert space. $\Phi_R^n$ are the recombination fluxes that enter the recombination channels and $\Phi_A^n$ are the injection fluxes that enter in the injection channels (acceptor side).

a finite bandwidth of $4J$. As $K$ tends to infinity, the term $K(J_{i,j})^2$ stabilizes to a constant value, converging towards $J^2$. This implies that each hopping integral $J_{i,j}$ diminishes as $K$ increases, ensuring that the bandwidth remains constant. $g$ signifies the electron-vibration coupling parameter (we used the same phonon mode for all sites, $g_i = g$). The Hamiltonian $H_R$ describes the recombination processes. As stated above these processes can be represented by a coupling to a continuum of states and in the formalism used here we just need to model the self-energy $\Delta_R(z)$ that represents this coupling to a continuum. We consider the cases where the continuum is a wide band ($\Delta_R(z) = -i\Gamma_R$) and narrow band as discussed in Section 5.2. All energies are expressed in units of $J$, which is typically of the order of 0.2 eV. All parameters are chosen to fall within a realistic experimental range [30–32, 41–44]. Finally we work in the limit where the temperature is zero, because band energies and phonon energies are well above the thermal energy at room temperature.

## 3 The scattering formalism

The injection process is analyzed within the complete Hilbert space including both electron and vibration modes, as illustrated in Figure (4).

The initial state $|I\rangle$ corresponds to the electron on the donor side with no excited phonon of the vibration mode. The hopping term $m$ enables transfer to the CTS $|0\rangle$ state with zero phonon. Then, the electron in the CTS $|0\rangle$ state couples with the phonons passing to the $|1\rangle, |2\rangle...|n\rangle$ states of the vertical chain. On the acceptor side or donor side, there is no more coupling with the phonons of the CTS.

### 3.1 Channels and probabilities

From a $|n\rangle$ state, there are two channels through which the electron leaves the CTS. The $nA$ channel corresponds to propagation on the acceptor side, and the $nR$ channel corresponds to the recombination process. The probabilities of injection into the $nA$ ($nR$) channels are $\Phi_A^n$ ($\Phi_R^n$). The probabilities $\Phi_A^n$ and $\Phi_R^n$ of injection into the $nA$ and $nR$ channels are calculated from scattering theory and depend only on the self-energies $\Delta_A(\varepsilon_I - n\hbar\omega)$ and $\Delta_R(\varepsilon_I - n\hbar\omega)$ that describe the corresponding channels. The quantity $\Delta_A(\varepsilon_I - n\hbar\omega)$ corresponds to the injection self-energy that depends on the model for the acceptor. In contrast, $\Delta_R(\varepsilon_I - n\hbar\omega)$ represents

the recombination self-energy. To calculate the self-energies $\Delta_A(z)$ due to the coupling with the acceptor, we use the recursion method coupled to the DMFT as shown in the reference [34].

It is important to determine the expressions of the probabilities $\Phi_A^n$ and $\Phi_R^n$ of electron injection and recombination over time along a given channel in Hilbert space. During the injection process for $t > 0$, considering the current $j_L(t)$ along a bind $L_n$ issued from the $|n\rangle$ of the vertical branch, we can write the probability formula $\Phi^n$:

$$\Phi^n = \int_0^{+\infty} j_L(t)dt = -\frac{1}{\pi}\int \text{Im}\,\sigma_n(z)\,|G_n(z)|^2\,dz, \tag{2}$$

where $G_n(z)$ is the amplitude of the matrix element of the Green's function $G(z) = \frac{1}{z-H}$ given by $G_n(z) = \langle I|G(z)|n\rangle$ (eq (12)). In our case it must be replaced by either $\Delta_A(z-n\hbar\omega)$ (coupling to the $nA$ channel) or $\Delta_R(z-n\hbar\omega)$ (coupling to the $nR$ channel) or $\Sigma_{n+1}(z)$ (coupling to the part of the vertical branch upper the site $|n\rangle$). Based on this formula, we can express the probabilities $\Phi_A^n$ ($\Phi_R^n$) injected into a right channel (left channel) from site $|n\rangle$ and the probabilities injected into the vertical channel $\Phi_V^{n+1}$ above site $|n\rangle$ on the vertical axis.

$$\Phi_A^n = -\frac{1}{\pi}\int Im(\Delta_A(z-n\hbar\omega))\,|G_n(z)|^2\,dz,$$

$$\Phi_R^n = -\frac{1}{\pi}\int Im(\Delta_R(z-n\hbar\omega))\,|G_n(z)|^2\,dz, \tag{3}$$

$$\Phi_V^{n+1} = -\frac{1}{\pi}\int Im(\Sigma_{n+1}(z))\,|G_n(z)|^2\,dz.$$

Here, $\Sigma_{n+1}(z) = (n+1)g^2 G_{n+1}^2(z)$ is the self-energy due to the coupling of site $|n\rangle$ with the upper part of the whole system above $|n\rangle$. From these fluxes, it is possible to express the quantum yield $Y$, which is defined by the probability of the electron being injected into the acceptor and the average vibration energy $E_{TS}$ on the CTS [45]. We obtain:

$$Y = \sum_{n=0}^{\infty}\Phi_A^n = 1 - \sum_{n=0}^{\infty}\Phi_R^n,$$

$$E_{TS} = \sum_n n\hbar\omega[\Phi_A^n + \Phi_R^n]. \tag{4}$$

In this work, the expressions for $Y$ and $E_{TS}$ are central for understanding what's happening at the donor-acceptor interface, as we'll see below.

## 3.2 Calculation of Green's functions and self-energies

We give now a few elements concerning the calculation of the Green's functions and self-energies. In the initial state $|I\rangle$, the electron has an on-site energy $\varepsilon_I$, and it is coupled to the zero-phonon injection state $|0\rangle$. Thus, the diagonal element of the Green's function at (I) is:

$$G_I(z) = \frac{1}{z - \varepsilon_I - m^2\tilde{G}_0(z)}, \tag{5}$$

where $\tilde{G}_0(z)$ is the diagonal element of Green's function on site 0 of a chain where site (I) has been removed (i.e. the chain starting at site 0 and this reasoning holds for all $\tilde{G}_n(z)$),

$$\tilde{G}_0(z) = \langle 0|G(z)|0\rangle. \tag{6}$$

On the CTS site $|0\rangle$, after deleting the site $|I\rangle$, Green's function is given by:

$$\tilde{G}_0(z) = \frac{1}{z - \varepsilon_0 - \Delta_R(z) - \Sigma_0(z) - \Delta_A(z)}, \tag{7}$$

where $\varepsilon_0$ is the energy of site $|0>$ and $\Sigma_0(z)$ represents the self-energy of the full vertical branch, which is given by:

$$\Sigma_0(z) = \cfrac{g^2}{z - \varepsilon_0 - \hbar\omega - \Delta_A(z - \hbar\omega) - \cfrac{2g^2}{z - \varepsilon_0 - 2\hbar\omega - \Delta_A(z - 2\hbar\omega) - \frac{3g^2}{\dots}}} = \cfrac{b_0^2}{z - a_1 - \cfrac{b_1^2}{z - a_2 - \frac{b_2^2}{\dots}}}. \qquad (8)$$

Starting from the initial site $|0\rangle$ in the vertical chain, we can calculate all Green's functions corresponding to each site of the vertical chain (the phonon mode). Therefore, we have:

$$G_1(z) = \langle 1|G|0\rangle = \tilde{G}_1 b(0)\tilde{G}_0, \qquad (9)$$

$$G_2(z) = \langle 2|G|0\rangle = \tilde{G}_2 b(1)\tilde{G}_1 b(0)\tilde{G}_0 = \langle 2|G|1\rangle\langle 1|G|0\rangle. \qquad (10)$$

The general formula becomes:

$$G_n(z) = \langle n|G|0\rangle = \frac{1}{b(n)}\prod_{p=0}^{n} b(p)\tilde{G}_p. \qquad (11)$$

Starting from the initial site $|I\rangle$, the first hopping parameter is $b(0) = m$, the second hopping parameter is $g$ etc... So the Green's function of a $|n\rangle$ state is given by:

$$G_n(z) = m\tilde{G}_I(z)\tilde{G}_0(z)\prod_{i=1}^{n} g\sqrt{i}\tilde{G}_i(z). \qquad (12)$$

$G_n(z)$ is the matrix element of Green's operator linking the initial state to the $n$ state, as shown in equation (12). Finally, we note that in the limit of small $m$ $|G_I(z)|^2$ behaves like a delta function which simplifies the equation (3) with integral expressions [34].

# 4 Analysis of the injection into a single band through the CTS

As explained in the previous section we treat injection in the acceptor band and recombination processes, formally on the same footing. In both cases, the electron is injected from the donor to the CTS and then to a continuum of states. This continuum represents either the acceptor band or the cascade of intermediate states which ultimately lead to the electron-hole recombination. Before analyzing situations where both processes are in competition it is interesting to pause on the concept of injection from the initial site $|I\rangle$ to the CTS, which is coupled to a single continuum of states. We shall differentiate between two extreme cases depending on whether the CTS is coupled to a narrow or a wide continuum of electron states, compared to other energy ranges. In particular we describe below models for the recombination that correspond to injection in a wide or narrow band.

## 4.1 Narrow-band limit

Let us consider first the case when CTS is coupled to a narrow spectrum continuum of states, compared to other energies. Initially, when the electron is on the donor site, if $m$ is small, the electron has a well-defined energy $\varepsilon_I$, while the vibrational modes are in their ground state without phonons (at zero temperature). Energy conservation dictates that the final state energy $E_F$ must satisfy $E_F = \varepsilon_I = n\hbar\omega + E_B$, where $E_B$ is in the band continuum (between $E_{min}$ and $E_{max}$) beyond the CTS, and $n\hbar\omega$ represents the energy of the $n$ phonons emitted on the CTS. If $\varepsilon_I$ is too high, the electron must first excite phonons on the CTS to reduce its energy to fit within the narrow band between $E_{min}$ and $E_{max}$. Consequently, as $\varepsilon_I$ increases, the number

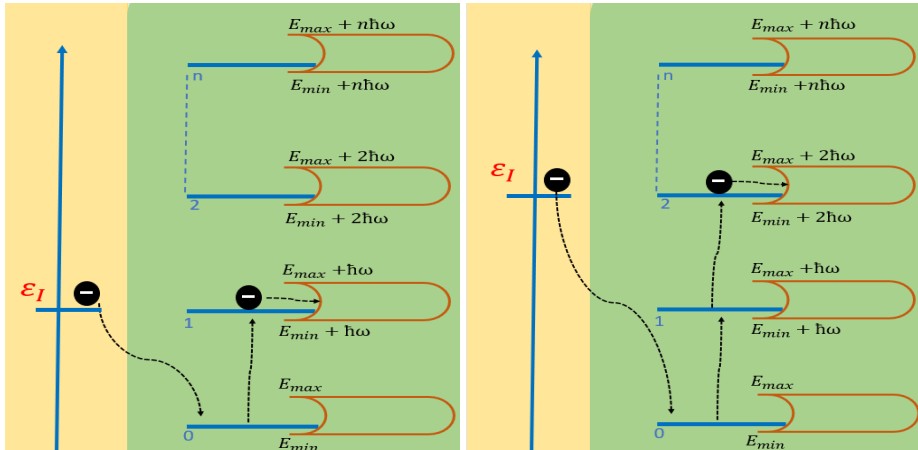

Figure 5: Illustration of the phonon emission process during electron injection: On the left, the electron must excite one phonon to be injected, and on the right it must excite two phonons to be injected.

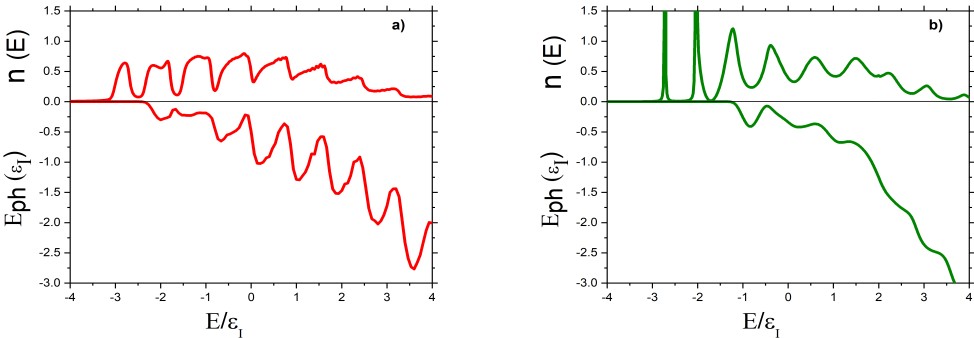

Figure 6: The spectral density $n(E)$ and the average energy lost in the CTS $E_{ph}(\varepsilon_I)$ ($E_{ph} = -E_{TS}$). a) N modes (Model A), b) 1 mode (Model B) for $g = \sqrt{2}$ and $\omega = 0.8$.

of phonons emitted on the CTS also increases, and the additional energy transferred to the CTS approximately equals the increase in $\varepsilon_I$. The illustration of the phonon emission process is shown in the Fig.5

Results are illustrated in Figure 6. The upper panels show the local density $n(E)$ at site $|0\rangle$, while the lower panels show the average energy $E_{ph} = -E_{TS}$ lost due to phonon emission on the CTS. Let us discuss first the panel $b$) for which a phonon mode is present only on the charge transfer site. For $\varepsilon_I$ between $E_{Min} = -2$ and $E_{Max} = 2$ (the continuum's lower and upper limits for the pure electronic spectrum), the system does not require phonon emission for electron injection into the acceptor side. However, if $\varepsilon_I > E_{Max}$, the electron must emit a phonon to be injected into the continuum. Similarly, for $\varepsilon_I > 3J$, two phonons are needed, resulting in a staircase-like curve. For $\varepsilon_I$ in the range $[E_{Max}, \infty]$, the average energy of the phonons is approximately $E_{TS} \approx (\varepsilon_I - E_{Max} + \hbar\omega)$. In the case of panel $a$) there is an electron-phonon coupling inside the acceptor. Therefore the acceptor band is polaronic and is more complex than in the previous case. Still we see that the same phenomena of single or multiple phonon excitation occurs, leading to a similar behavior for the energy exchanged between the electron and the vibration mode of the CTS.

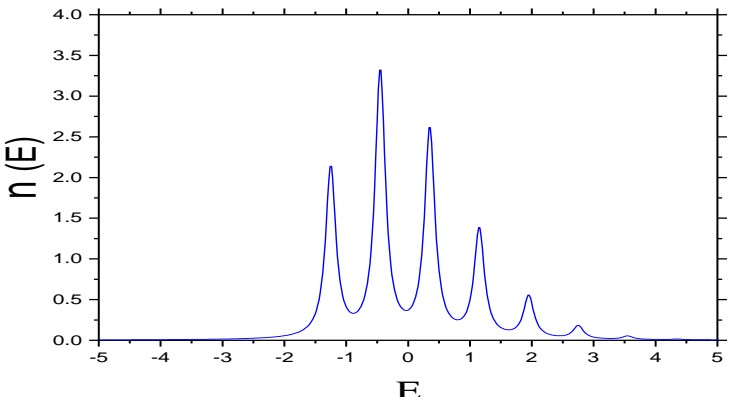

Figure 7: The spectral density on the ground state $|0>$ at 0 phonon in a wide band with: $g = 1$, $\omega = 0.8$ and $\epsilon = 0.1$.

## 4.2 Wide band limit

When considering charge injection into a wide band, it is useful to simulate this scenario by using a constant self-energy of the form $\Delta = -i\epsilon$ for the coupling of the CTS with the continuum. In the limit of small $m$, the scattering state for an initial energy $E$ is described as:

$$|\psi\rangle = \frac{1}{E - H}|0\rangle,$$

where $|0\rangle$ represents the ground state of the CTS without an electron-vibration coupling (i.e. for the Hamiltonian $H_0 = \omega a_0^\dagger a_0$), and $H$ is the Hamiltonian of the CTS with an electron-vibration coupling:

$$H = \omega a_0^\dagger a_0 + g(a_0^\dagger + a_0) - i\epsilon = \omega(a^\dagger a - \alpha^2) - i\epsilon.$$

Here, $a^\dagger = a_0^\dagger + \alpha$, $a = a_0 + \alpha$, and $\alpha = \frac{g}{\omega}$, resulting in a displaced and damped harmonic oscillator. The eigenvalues of $H$ are:

$$E_n = \omega(n - \alpha^2) - i\epsilon.$$

For the wavefunction $|\psi_n\rangle$ of the eigenstate of $H$, the projection onto the ground state $|0\rangle$ of the initial harmonic oscillator is:

$$P_n = |\langle \psi_n | 0 \rangle|^2 = \frac{\alpha^{2n}}{n!} e^{-\alpha^2}.$$

Figure 7 shows the spectral density on the ground state of the harmonic oscillator with 0 phonons. Peaks corresponding to the eigenenergies $E_n = \omega(n - \alpha^2) - i\epsilon$ of the Hamiltonian $H$ are observed. These peaks are broadened by the damping term $\varepsilon$ and have a weight $P_n$. The average number of phonons emitted, which we are interested in, is given by:

$$\omega N_{\text{ph}}(E) = \frac{\langle \psi | H_0 | \psi \rangle}{\langle \psi | \psi \rangle},$$

where $H_0$ is expressed as:

$$H_0 = \omega a_0^\dagger a_0 = H - g(a^\dagger + a) + 2g\alpha + i\epsilon.$$

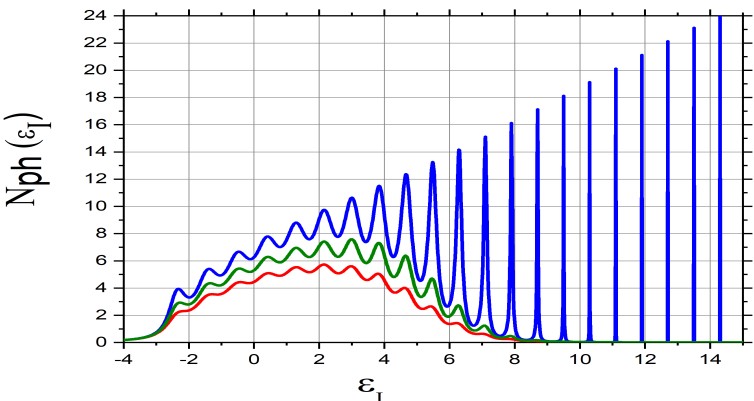

Figure 8: Average number of phonons ($N_{ph}$) as a function of injection energy $\varepsilon_I$ for a coupling $g = \sqrt{2}$, a frequency $\omega = 0.8$, and different damping values: $\epsilon = 1.25 \times 10^{-6}$ for the blue curve, $\epsilon = 0.2$ for the green curve, and $\epsilon = 0.3$ for the red curve.

Substituting $H_0$ into the expression for the number of phonons, we obtain:

$$\omega N_{\text{ph}}(E) = 2g\alpha \sum_n Q_n(E)\left(E_n + i\epsilon - 2g\Re\left(\frac{E - E_n}{E - E_{n+1}}\right)\right),$$

with

$$Q_n(E) = \frac{P_n/|E - E_n|^2}{\sum_r P_r/|E - E_r|^2}.$$

In figure 8, the average number of phonons is plotted as a function of energy $\varepsilon_I$ for different values of $\epsilon$ ($\Delta = -i\epsilon$). As $\epsilon$ approaches zero, the average number of phonons exhibits peaks at $\varepsilon_I = E_n$, with:

$$\omega N_{ph}(E \to E_n) = E_n + 2g\alpha.$$

This indicates that at these injection energies and for small $\epsilon$, the populated level is $|\psi_n\rangle$, containing $E_n + 2g\alpha$ phonons. As $\epsilon$ increases, the peaks diminish, and the phonon distribution smooths out. This suggests that coupling to broadband reduces the effect of discrete energy states, leading to a more continuous phonon distribution. At high $\varepsilon_I$, the scattering state approximates the ground state $|0\rangle$ of the initial harmonic oscillator, leading to a number of emitted phonons that are approaching zero. These observations can be interpreted using the concept of Wigner time, which is the time required to tunnel through a potential barrier [46–48]. As $\varepsilon_I$ increases, the Wigner time grows, causing the electron to be injected into the continuum, within a typical time $\hbar/\epsilon$, before transferring excess energy to CTS vibrations. Thus, at higher $\varepsilon_I$, the electron has insufficient time to excite phonons before moving to the continuum.

# 5 Numerical study

In this part, we analyze complex situations with all the possible effects included in the embedded model such as electron-vibration coupling, electrostatic potential, the role of the environment, role of the type of recombination process. We start by analyzing the electronic structure through the density of states (DOS) on the CTS $|0>$. This part is independent of the

recombination which has a negligible influence on the DOS. Then we analyze the yield and energy transfer on the CTS. We consider first a few cases for some representative parameter sets. We continue with a presentation of the phase diagram as a function of two essential parameters which are the injection energy and the recombination rate. These phase diagrams confirm the importance of the nature of the dominant recombination process for the yield and the energy transfer on the CTS. Note that the parameters of the Hamiltonian are given in units of $J$ which is typically of the order of 0.2 eV. All parameters are chosen such that they are in a realistic experimental range to model the prototypical PCMB and C60 acceptor systems [30–32,41–44].These molecular solids have electronic bands which are narrow (typically 4J which is less than one eV) and have molecular vibration modes of high frequency which can be of the order of 0.1/0.2 eV.

## 5.1 Electronic structure: Role of environment and electrostatic potential

We study the impact of the environment beyond the CTS and compare two types of environments. The first model involves a Bethe lattice coupled to $N$ phonon modes (Model A), meaning that each site has a phonon mode to which the electron is coupled when occupying the site, while the second model consists of a simple Bethe lattice (Model B). These two models can be seen as two extreme cases of electron-phonon coupling heterogeneity and allow for the analysis of the specific role of the CTS among all the sites of the acceptor. We compare these two models in three typical cases of electrostatic potential: Zero potential, repulsive potential and attractive potential. Figure 9 shows the spectral density on the CTS for parameters $g = 1$, $\omega = 0.8$, and $\Gamma_R = 0.1$. The figure consists of three panels: With no potential, with a repulsive potential ($V = 2$), and with an attractive potential ($V = -2$). Model A is depicted in red, while model B is shown in green. For zero potential $V = 0$, as shown in Figure 9 a), the spectral density exhibits oscillations with maxima approximately separated by the phonon energy $\omega = 0.8$, which is characteristic of polaronic bands. Model A shows more pronounced oscillations, but both models display similar trends. With a repulsive potential $V = 2$, as depicted in Figure 9 b), the spectral density shifts to higher energies in both models. Model B is more sensitive to this shift, yet the overall trends remain comparable.

In the case of an attractive potential $V = -2$, Figure 9 c) shows that the density of states presents localized peaks at the band bottom. Model B is again more sensitive, although the general trends are consistent. Note that the total weight of the localized on the state $|0 >$ can be quite high of the order of $0.3-0.4$ for $g = 1$. These states are analogous to bound electronic states in atoms or impurity states in semiconductors and are geometrically localized around the CTS. Although an infinite series of such states might be expected near the minimum energy of the continuum, their weights are too low to be detected numerically.

Now, let's focus on the ground state of the polaron in the presence of an attractive potential, which corresponds to a localized polaronic state (Model A). Figure 10 a) shows us the distribution of the wave function weight on the different layers of the lattice for an attractive potential $V = -2$ as a function of coupling g. Regarding the weight on the central site, that is the CTS, (black curve), we first notice that as g tends towards zero, the weight tends towards a limit, which is that of the non-interacting localized state. We also observe that the weight on the other layers decreases as the coupling increases. When it becomes very significant, the wave function is entirely localized on the central site. In Figure 10 b), we now show the average number of phonons on the different layers of the lattice, again for an attractive potential $V = -2$, as a function of coupling g. We notice that except for the central site, the average number of phonons is extremely low on the other layers of the lattice. If we relate this to the weights of the wave function, we can say that the majority of the weight on the layers comes from the weight at zero phonons, except for the CTS (layer zero).

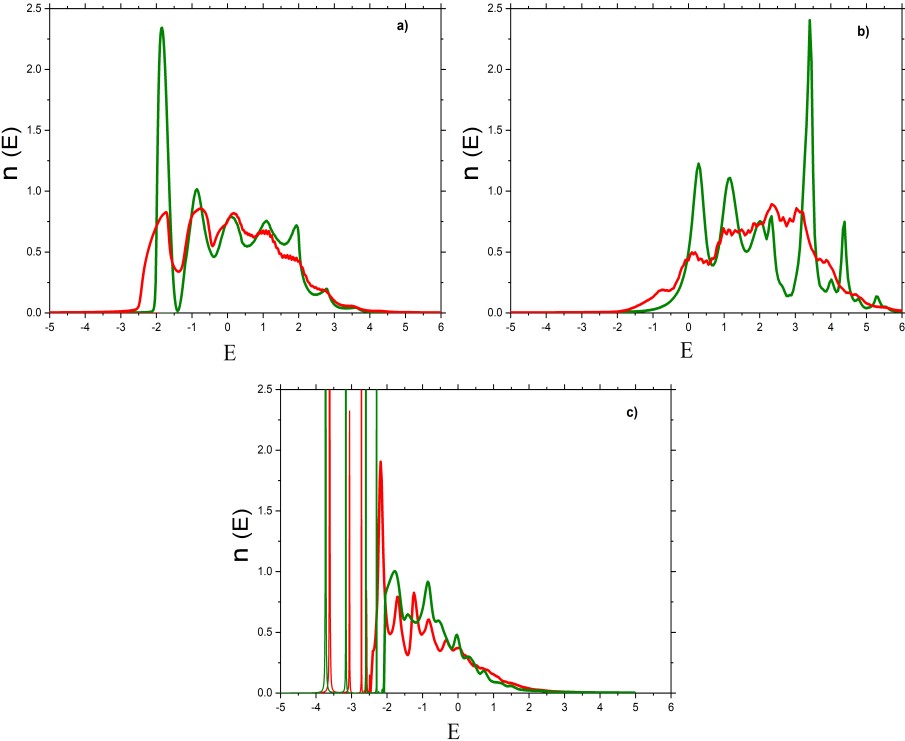

Figure 9: The spectral density on the CTS with parameters $g = 1$, $\omega = 0.8$, $\Gamma_R = 0.1$ a) in the absence of potential, b) $V = 2$ and c) $V = -2$. The red color corresponds to model A, and the green color represents model B.

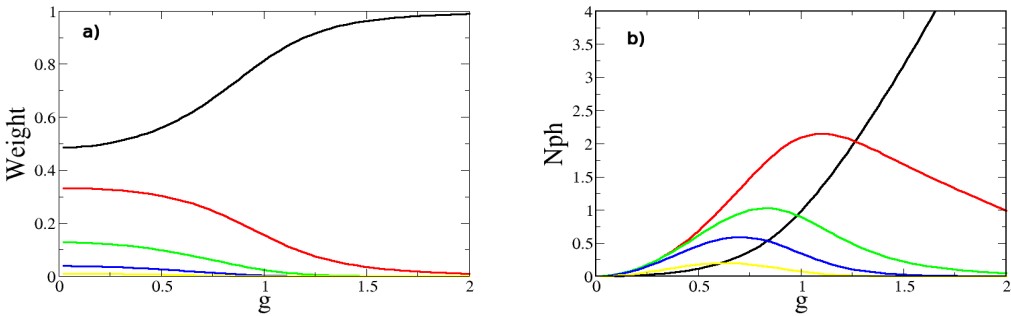

Figure 10: a) Weight of the wave function for different layers of the Bethe lattice and an attractive Coulomb potential $V = -2$ as a function of coupling g. b) Average number of phonons on the different layers of the Bethe lattice for an attractive Coulomb potential $V = -2$ as a function of coupling g, (red curve layer 1 *20, green curve layer 2 *100, blue curve layer 3 *400, and yellow curve layer 4 *800.

## 5.2 Quantum yield and energy transfer on the CTS

Figure 11 presents the quantum yield and the average phonon energy for parameters $g = 1$, $\omega = 0.8$, and $\Gamma_R = 0.1$. The figure includes three panels: With no potential, with a repulsive potential ($V = 2$), and with an attractive potential ($V = -2$). Model A is shown in red, and model B is shown in green. The recombination is treated in this case as injection in a wide band ($\Delta_R = -i\Gamma_R$).

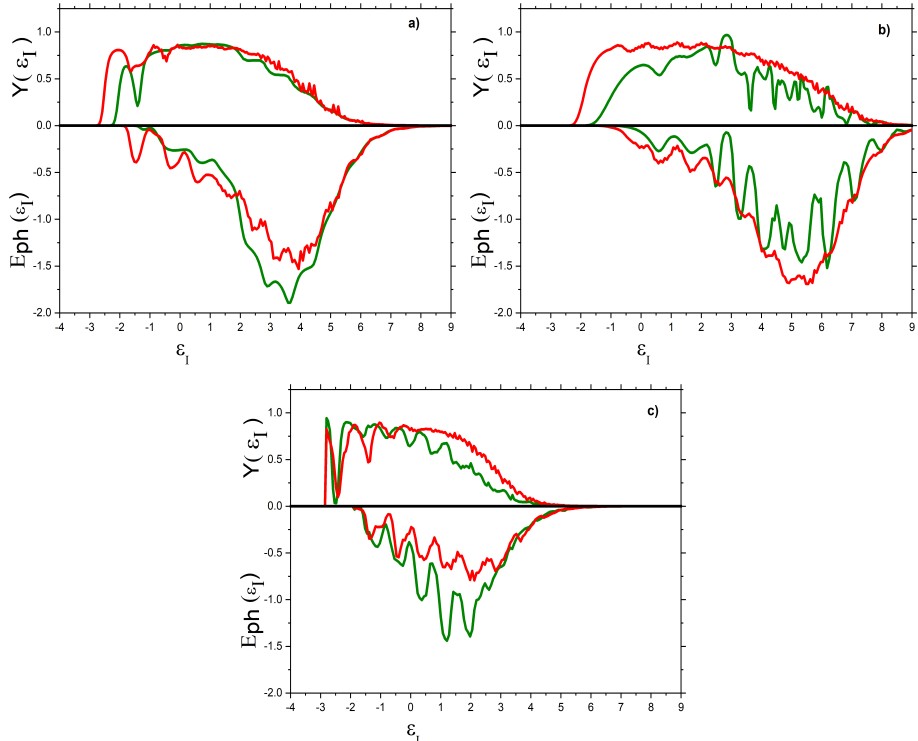

Figure 11: The quantum yield, and the average phonon energy with parameters $g = 1$, $\omega = 0.8$, and $\Gamma_R = 0.1$ (a) in the absence of potential, (b) $V = 2$ and (c) $V = -2$. Red represents model A, and green represents model B.

We first observe that both models A and B yield similar results, showing that electron-phonon coupling beyond the CTS has a moderate effect. Figure 11 indicates that the average number of emitted phonons stays relatively small within the electronic band, with a yield close to one. We observe also that when going from attractive to repulsive electrostatic potential the region of high yield and high number of emitted phonons tend to increase. Yet in the case of attractive potential, the existence of a localized polaronic state close to the interface (see previous section 5.1) does not hamper an efficient injection, inside the acceptor band.

Finally as the initial electron energy $\varepsilon_I$ surpasses $E_{max} \simeq 2$, phonon emission increases and then decreases, while the yield decreases and tends to zero at high initial energies $\varepsilon_I$. This energy $E_{max}$ is slightly reduced for the attractive potential but increases to approximately $E_{max} \approx 3$ for the repulsive potential. This behavior is interpreted, as a competition between injection in the 'narrow' band of the acceptor and in the wide band that represents the recombination process. Indeed for $\varepsilon_I > E_{max} \simeq 2$ the CTS acts like a potential barrier as discussed previously (section 4.2). The Wigner time that is needed to pass the barrier increases with $\varepsilon_I$ and when it becomes larger than the recombination lifetime the recombination dominates. Before it recombines, the electron does not have enough time to emit phonons, and the yield and number of emitted phonons tend to be zero.

For all the results presented afterward, we use only model A with parameters $g = 1$ and $\omega = 0.8$. For the case of recombination from the acceptor treated as injection in a narrow band, we choose the self-energy of recombination $\Delta_R^n(z)$ at site $n$ of the vertical chain as:

$$\Delta_R^n(z) = \frac{m_1^2}{z - n\omega - \dfrac{J^2}{z - n\omega - \frac{J^2}{\cdots}}} \, .$$

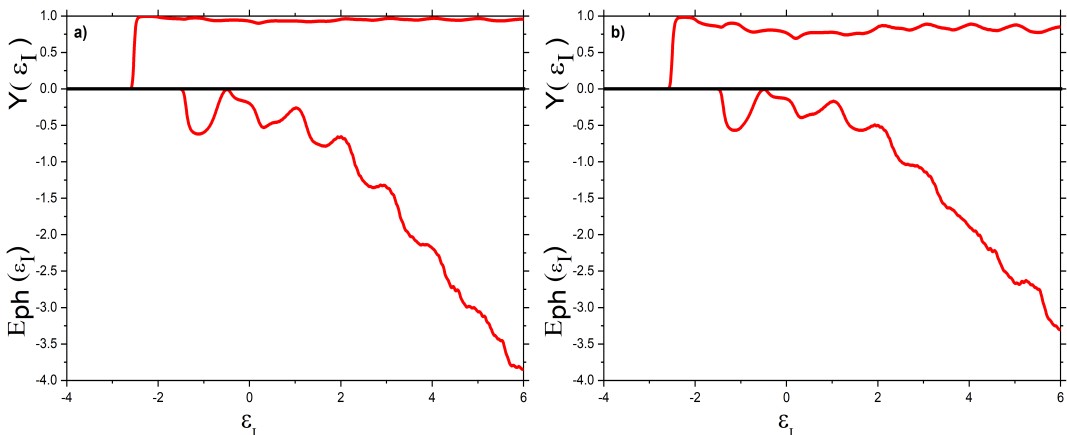

Figure 12: The average phonon energy and quantum efficiency with recombination in a narrow band. a) $m_1 = 0.25$ and b) $m_1 = 0.5$.

In figure 12, we present the average phonon energy and quantum efficiency with recombination in a narrow band for $g = 1$ and $\omega = 0.8$. a) $m_1 = 0.25$, and b) $m_1 = 0.5$. For this recombination model, both bands (environment side and recombination side) can roughly be viewed as a single narrow band. Therefore, we expect the number of phonons emitted on the CTS to increase with $\varepsilon_I$, which is indeed observed. Moreover, the ratio of injection rates into the environment and toward recombination is roughly given by the ratio of injections into a band representing the environment and a band representing recombination. This ratio depends little on the number of emitted phonons, resulting in a weak dependence of the quantum yield on the injection energy. This type of behavior is evident in both panels of Figure 12. Thus, we see that this type of recombination, modeled by injection in a narrow band leads to very different behavior compared to recombination with a wide band.

Let's now focus on the final case, which is recombination from the donor. For the electron on the donor side, there is competition between recombination at the donor site and injection on the acceptor side. In this scenario, the donor-acceptor hopping integral, $m$, plays an important role since the injection rate on the acceptor side is proportional to the square of $m$, according to Fermi's golden rule.

There are only two channels connected to the donor site. The first is linked to the acceptor via the hopping integral $m$, and the second is the recombination channel with the ratio $\Gamma_G$ (Figure 3). In figure 13, a) shows the quantum yield and the average phonon energy, while b) displays the average number of phonons emitted per injected electron. In figure 13 a), we observe a behavior similar to the case of recombination in a wide band on the acceptor side, which can be simply interpreted. As the injection energy increases, the density of states on the CTS at this energy $\varepsilon_I$ decreases, leading to a reduced injection rate into the acceptor. This allows recombination on the donor site, which lowers the quantum yield. In contrast, figure 13 b) shows a linear increase in the number of phonons emitted per injected electron. This is explained by the fact that once the electron is injected on the acceptor side, it cannot recombine at the donor site. This is because after emitting one or several phonons the energy left to the electron is no longer resonant with the initial donor site energy $\varepsilon_I$. Therefore, the electron must 'climb' the phonon excitation chain, in order to loose enough energy to be in resonance with the polaronic band of the acceptor.

In conclusion, after comparing the results of density, quantum yield, and average number of phonons with recombination for the two models A and B, whether in the presence of a null, attractive, or repulsive potential, we show that both models yield similar results. This sug-



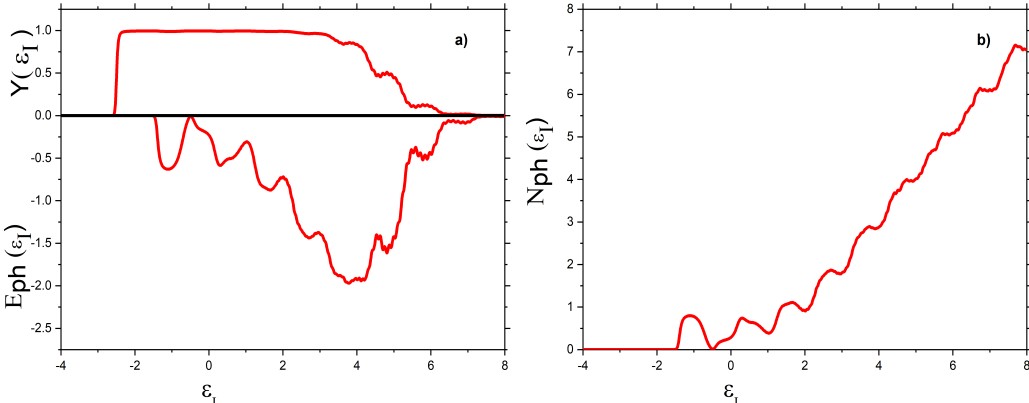

Figure 13: a) The average phonon energy and the quantum yield. b) The average number of phonons emitted per electron injected. Parameters are $g = 1, \omega = 0.8, m = 0.25$ and $\Gamma_G = 0.001$.

gests that the effect of the interface between the donor site and the acceptor site CTS plays a predominant role in the injection process, while the environment beyond the CTS does not significantly affect injection or the relevant physical quantities.

We find also that for attractive potential the existence of localized electron-hole bound states on the acceptor side does not hamper a good injection at higher energies in the acceptor band. Yet the electrostatic potential has some effect: When it passes from attractive to repulsive the energy window for high yield increases and the number of emitted phonons increases also. Finally, the type of recombination that dominates has a strong influence on the injection process.

## 5.3 Phase diagrams

The results presented just above are now analyzed more systematically on the basis of phase diagrams, shown for model A only. These diagrams represent the average number of phonons emitted on the CTS and the yield as functions of electron injection energy and recombination parameters. We define the "hot transfer regime" as emitting more than one phonon and the "cold transfer regime" as emitting fewer than one phonon. A "high-yield zone" is where quantum efficiency exceeds 0.5, and a "low-yield zone" is where it falls below 0.5. Figure 14 shows recombination from the acceptor in a wide band. The presence or absence of potential does not alter the overall diagram appearance. A finite hot transfer regime narrows with increasing recombination until it is replaced by the cold region. The upper part features a low-yield zone where both injection energy and recombination rate are high. While most of the hot transfer zone falls within the high-yield region, there is a small coexistence area between hot transfer and low yield. The potential primarily affects the quantitative aspects; repulsive potentials expand the hot transfer and high-yield regions, while attractive potentials reduce them. Figure 15 depicts recombination from the acceptor in a narrow band. Notably, there is no low-yield region. Injection into two narrow bands behaves similarly to injection into a single narrow band, resulting in minimal variations in yield. The average number of phonons increases linearly, creating a hot transfer region that spans up to $E_{\max}$. This behavior differs significantly from the wide-band recombination case. Figure 16 illustrates recombination on the donor side. Here, the hot transfer region covers the entire upper part of the diagram, resembling narrow-band recombination and differing from wide-band recombination on the acceptor side. Electrons injected into the CTS cannot return to the donor side for recombination (assuming very small $m$). The number of emitted phonons follows the behav-

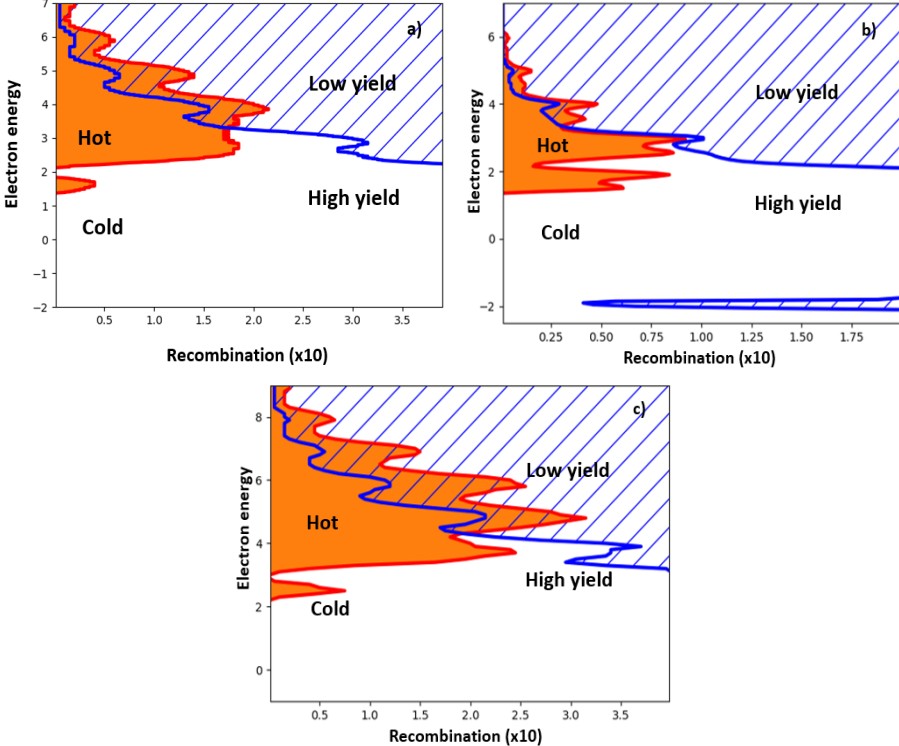

Figure 14: Phase diagrams for the recombination mechanism in a wide band are shown. The parameters are the electron injection energy and the recombination rate $\Gamma_R$. a) The case without potential, b) the case with an attractive potential, and c) the case with a repulsive potential.

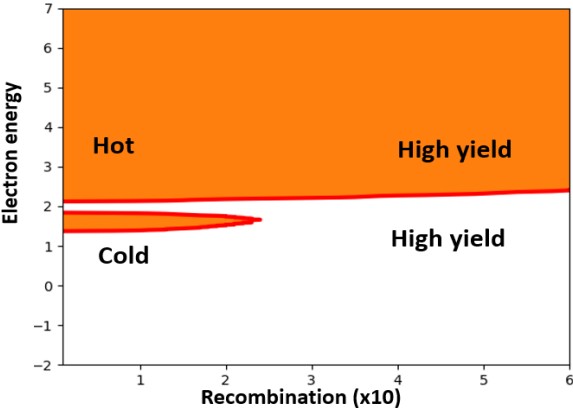

Figure 15: Phase diagram for recombination in a narrow band. The parameters are the electron injection energy and the recombination rate, described here by the parameter $m_1$. The electrostatic potential is zero.

ior observed for no recombination, while the yield mimics the wide-band case. Increased $\varepsilon_I$ extends the injection time to the acceptor side, allowing more time for donor-side recombination. In conclusion, these phase diagrams highlight the critical role of recombination type at the donor-acceptor interface in determining the quantum yield of an organic cell and the presence of hot charge transfer states. This aspect, less discussed in the literature, represents a key finding of this study.

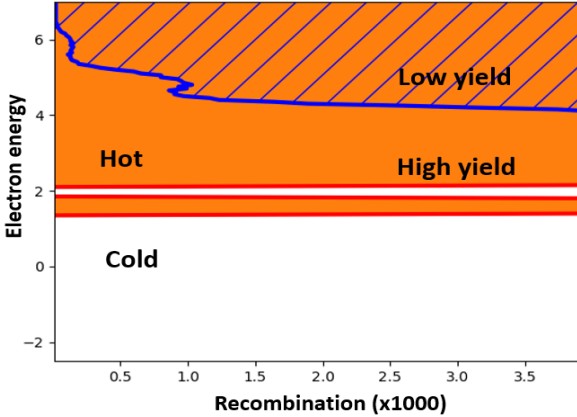

Figure 16: Phase diagram for recombination on the donor side in a wide band, with no electrostatic potential. The parameters are the electron injection energy and the recombination rate $\Gamma_G$. The parameters are the same as in the previous figure. To represent this figure, we use the number of phonons emitted per electron passing through the CTS, i.e., per electron injected on the acceptor side.

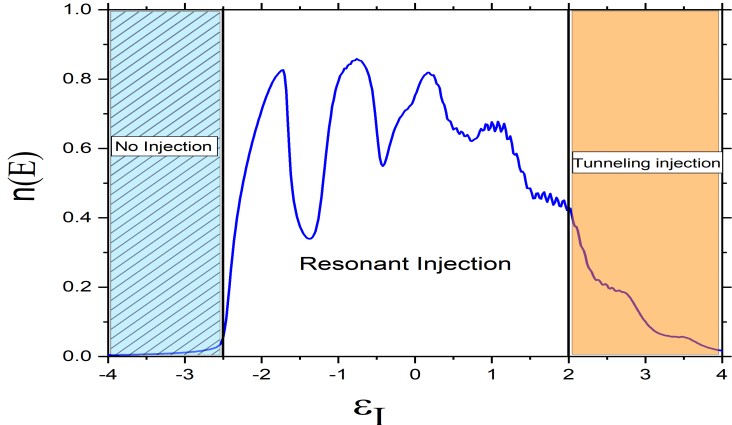

Figure 17: Charge transfer regimes based on the electron injection energy $\varepsilon_I$. The characteristics of these regimes are discussed in the text.

## 6   Discussion

Within the framework of the embedded CTS model and for the range of parameters considered here, we ultimately distinguish three main regimes based on the value of the injection energy $\varepsilon_I$. In order of increasing $\varepsilon_I$, we find a no-injection regime, a resonant injection regime, and a tunneling injection regime. We summarize all the results in Figure 17. At low energy, $\varepsilon_I$ lies below the continuum of states, and injection is not possible. This is valid at zero temperature, which is one of the assumptions of our study. At finite temperatures, thermally activated processes may exist. In a higher energy range, the injection is resonant. Indeed $\varepsilon_I$ lies within the continuum of states, allowing quick injection from the donor to the CTS and then rapid electron transfer to the acceptor without phonon excitation on the CTS (cold CTS). We found above that a repulsive potential seems more favorable to a good injection with a cold CTS than an attractive potential, even though the electrostatic potential does not qualitatively alter the injection behavior. Notably, even if an attractive potential can create a bound state with a significant weight on the CTS (> 0.3-0.4), it does not prevent efficient injection into

the continuum of states. Thus, the existence of bound states created by the electron-hole interaction in the acceptor is not incompatible with good injection and high quantum yield. In this resonant regime, the effect of the environment beyond the CTS is also moderate.

In an even higher energy range, we find a tunneling regime. For the electron to be injected into the environment beyond the CTS, its initial energy $\varepsilon_I$ must first be reduced by exciting phonons on the CTS to get in resonance with the acceptor band. This situation is analogous to the crossing of a potential barrier, and the concept of Wigner time for barrier crossing can be applied here. It indicates that in this regime the electron spends more time on the CTS and can therefore be more sensitive to recombination. Additionally, as the density of states on the CTS decreases at high energies $\varepsilon_I$, the injection time from the donor increases, according to Fermi's golden rule. Thus, in this tunneling regime, the residence times of the electron on the donor side and on the CTS are longer. Therefore if recombination processes are fast enough, they will reduce the quantum yield of the injection and possibly lead to a cold CTS. In the wide band recombination model the time for recombination is constant and will always become smaller than the Wigner time for residence on the CTS when $\varepsilon_I$ increases. Yet in the narrow-band recombination model, the recombination time also increases with $\varepsilon_I$. Therefore, once injected into the CTS, the electron may still have a significant injection yield on the acceptor side (see Figure 11 b)).

# 7    Conclusion

In this article, we have developed and studied a charge injection model at a donor-acceptor interface, typically applicable in organic semiconductors. This model, which we refer to as the "embedded charge transfer state model", allows for the analysis of the injection process by focusing on the electron transfer to the CTS site and simplifying the environment on both the acceptor and donor sides. This model accounts for electron-lattice interactions, electrostatic potential on the acceptor side, and geminate recombination between electron and hole. We have employed a framework based on dynamic mean-field theory and scattering theory, which provides us with an exact numerical solution of this model.

This study shows that the coupling between electrons and vibrational modes affects quantum efficiency, introducing the concepts of hot and cold charge transfer states. Our findings indicate that interface parameters are crucial for optimizing cell performance and that the interface's impact is more significant than environmental variations beyond the CTS. In addition, the electrostatic potential can play a moderate role in the injection process, even when it is attractive and creates localized bound electron-hole states at the interface. The charge injection process is described as a transfer into a continuum at an energy $\varepsilon_I$. When $\varepsilon_I$, is in the bottom part of the continuum an efficient resonant injection regime occurs with a cold CTS. At higher values of $\varepsilon_I$ the electron must first loose energy by emitting phonons before going in the acceptor side. In this tunneling regime the electron spends more time on the donor orbital and then on the CTS site and the recombination can be favored.

Although a detailed comparison with a given system is beyond the scope of this work, we believe that the present model offers a simplified but comprehensive approach to advanced analyses of experimental results [11]. It should be helpful for a better understanding of the exciton dissociation process and for finding ways to improve the organic solar cells. The present embedded charge transfer state model could still be improved by considering several orbitals and several modes per site and even finite temperature. We emphasize that the modeling of the dynamics of the recombination process is also an essential ingredient.

## Acknowledgments

The numerical calculations have been performed at Institut Néel, Grenoble and Condensed Matter Physics Laboratory, Tunisia. We would like to express our gratitude to Gabriele d'Avino for discussions about the choice of parameters for the model. We also thank Jean-Pierre Julien and Nicolas Cavassilas for their insightful discussions.

**Funding information** A. Perrin thanks France 2030 ANR QuantForm-UGA for PhD Grant.

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
