# Peer review of "Exciton dissociation in organic solar cells: An embedded charge transfer state model"

_SciPost Physics, doi:SciPost Phys. 18, 038 (2025)_

## Round 1 · Referee Report · Anonymous (Referee 1) · 2024-10-17

Report
Requested changes
1-This paper can be regarded as a follow-up paper to Ref. 38, while providing more detailed analyses. The authors should address this point in the introduction and clarify the differences and novelty compared to Ref. 38.
2-In Eq. (1), the expression of H_R is not presented. If the recombination is included as a simple self-energy, that should be written explicitly.
3-For better readability, it may be helpful to explain that the sign of V characterizes the electrostatic potential in Section 2.4.
4-In Section 2.4, the authors state "We work in the limit where the temperature is zero, because band energies and phonon energies are well above the thermal energy at room temperature." How can the omission of low-frequency vibrations be justified?
5-The recombination rate \Gamma_R is not defined. The authors must define it clearly.
6-How many basis sets are used for vibrations?
7-Why does the axis in Fig. 5 point downward?
8-It would be helpful to explain why the density of states presents the localized peaks in Fig. 9(c).
9-Which model is used for Fig (10)?
10-In Fig. (17), the label of the y-axis is missing.
Recommendation
Ask for minor revision
Attached are the answers to the questions, numbered according to the corresponding questions: 1- Indeed, we studied the model in its simplest form in reference [34] (The reference 38 has become 34). We explained in the introduction (page 3) what new developments we presented in this work by adding the following paragraph: “A first analysis of charge injection was presented in our previous work \cite{chika2022model}. Here we enlarge our approach by considering several types of recombination processes, various type of environments, attractive or repulsive potential and by analyzing more deeply the process of injection and its interdependence with phonon emission on the CTS”. This addition highlights the differences and the extended scope of our current work compared to Ref. 34, as suggested.
2- The Hamiltonian HR describes the recombination processes. As stated above these processes can be represented by a coupling to a continuum of states and in the formalism used here we just need to model the self-energy ∆R(z) that represents this coupling to a continuum. We consider the cases where the continuum is a wide band (∆R(z) = −iΓR) and narrow band as discussed in Section 5.2. All energies are expressed in units of J, which is typically of the order of 0.2 eV. All parameters are chosen to fall within a realistic experimental range [30–32, 41–44].
3- The sign of V characterizes the sign of the electrostatic potential: if V>0 then the potential is repulsive and if V<0 then the potential is attractive. This has been added to the text in the last paragraph (page 5).
4- We have added a sentence in the introduction in section 2.4.”The Hamiltonian H of our model takes into account a tight-binding hopping model for the electron and includes the essential coupling with local vibration modes, as is common for molecular solids [21].”
5- This definition of Gamma_R was clarified on page 6 at the end of section 2.4.
6- In this model, we consider one orbital per site, and for each phonon mode, we account for coupling with multiple phonons. Typically we take into account at least the emission of ten phonons which is largely enough here to get a good convergence of the computation.
7- Thank you for your remark. It's a mistake; we have corrected the figure.
8- As explained now “These states are analogous to bound electronic states in atoms or impurity states in semiconductors and are geometrically localized around the CTS. Although an infinite series of such states might be expected near the minimum energy of the continuum, their weights are too low to be detected numerically.”
9- In Figure 10, we present the results for the N phonon modes (Model A).We added “Model A” in the sentence just before the figure.
10- You're right, the y-axis is the spectral density n(E).This is now shown.

Author: Khouloud Chika on 2024-11-12 [id 4958]
(in reply to Report 2 on 2024-10-18)Attached are the answers to the questions, numbered according to the corresponding questions:
1- We have addressed this point at the end of Section 2 with the following sentence “All energies are expressed in units of $J$, which is typically of the order of $0.2$ eV. All parameters are chosen to fall within a realistic experimental range [30–32, 41–44]“ and at the beginning of section 5 with the sentence “Note that the parameters of the Hamiltonian are given in units of J which is typically of the order of 0.2 eV. All are chosen such that they are in a realistic experimental range to model the prototypical PCMB and C60 acceptor systems [30–32, 41–44]”
2- This remark is connected to the previous one. We have addressed this point in Section 5 by adding the following paragraph: “Note that the parameters of the Hamiltonian are given in units of $J$ which is typically of the order of $0.2$ eV. All are chosen such that they are in a realistic experimental range to model the prototypical PCMB and C60 acceptor systems \cite{zheng2019charge,d2016electrostatic,faber2011electron,castet2014charge,antropov1993phonons,d2014electronic,richler2019influence}. These molecular solids have electronic bands which are narrow (typically 4J which is less than one eV) and have molecular vibration modes of high frequency which can be of the order of 0.1/0.2 eV. ”
3- We appreciate the reviewer's suggestion regarding the need for experimental comparisons. In the revised manuscript, we emphasize that in the conclusion (last paragraph) “Although a detailed comparison with a given system is beyond the scope of this work, we believe that the present model offers a simplified but comprehensive approach to advanced analyses of experimental results [11]. It should be helpful for a better understanding of the exciton dissociation process and for finding ways to improve the organic solar cells. The present embedded charge transfer state model could still be improved by considering several orbitals and several modes per site and even finite temperature. We emphasize that the modeling of the dynamics of the recombination process is also an essential ingredient.”

---

## Round 1 · Referee Report · Anonymous (Referee 2) · 2024-10-18

Strengths
Weaknesses
Report
Requested changes
- The units of most physical quantities are unclear. Authors stated all energies are expressed in units of J. The J is the hopping matrix elements of a carrier between nearest neighbors. But typical value of J is not given in the paper.
- The values of parameters are very important for the results and conclusions. For most calculations, the following parameters are used, g=1 and sqrt(2); w=0.8; eps=1.25x10^(-6), 0.1, 0.2, 0.3; V=-2, 2. But authors didn’t explain reasonability of these values. However, the values g=1 and w=0.8 seems too large for it means energy level of a phonon is similar to a carrier.
- The paper is theoretical work, all results and conclusions are not compared with experiments. At least, the authors should give some discussions about possible methods to check the results and conclusions.
Recommendation
Ask for minor revision

---

## Round 2 · Referee Report · Anonymous (Referee 1) · 2024-11-26

Report

The authors have carefully revised the paper in terms of reviewing report, and properly reply all questions advanced by referees. The revised version meets criteria of SciPost, I am pleased to recommend acceptation of the paper for publication.

Recommendation

Publish (meets expectations and criteria for this Journal)

---

## Round 2 · Referee Report · Anonymous (Referee 2) · 2024-11-28

Report

I'm pleased to see that the authors have carefully revised the paper. The current version is suitable for publication in SciPost Physics.

Recommendation

Publish (meets expectations and criteria for this Journal)

---

## Round 2 · List of Changes

- We've added paragraph 5 to the introduction, as follows: A first analysis of charge injection was presented in our previous work \cite{chika2022model}. Here we enlarge our approach by considering several types of recombination processes, various type of environments, attractive or repulsive potential and by analysing more deeply the process of injection and its interdependence with phonon emission on the CTS

-We've added paragraph 2 to the section 2.3, as follows: Several mechanisms can lead to the genimate recombination between the electron in the acceptor side and the hole left in the donor side. This can be for example photon emission or multiple emission of low energy phonons which lead to a transfer of the electron into localized states near the interface and then to recombination with the hole [14].

-We've added paragraph 1 to the section 2.4, as follows: The Hamiltonian $H$ of our model takes into account a tight-binding hopping model for the electron and includes the essential coupling with local vibration modes, as is common for molecular solids \cite{bredas2004charge}

-We've added the final paragraph to section 2.4, as follows: As stated above these processes can be represented by a coupling to a continuum of states and in the formalism used here we just need to model the self-energy $\Delta_R (z)$ that represents this coupling to a continuum. We consider the cases where the continuum is a wide band ($ \Delta_R(z) = -i \Gamma_R $) and narrow band as discussed in Section 5.2. All energies are expressed in units of $J$, which is typically of the order of $0.2$ eV. All parameters are chosen to fall within a realistic experimental range \cite{zheng2019charge,d2016electrostatic,faber2011electron,castet2014charge,antropov1993phonons,d2014electronic,richler2019influence}.

-We've added the final paragraph to section 5, as follows: Note that the parameters of the Hamiltonian are given in units of $J$ which is typically of the order of $0.2$ eV. All parameters are chosen such that they are in a realistic experimental range to model the prototypical PCMB and C60 acceptor systems \cite{zheng2019charge,d2016electrostatic,faber2011electron,castet2014charge,antropov1993phonons,d2014electronic,richler2019influence}.These molecular solids have electronic bands which are narrow (typically 4J which is less than one eV) and have molecular vibration modes of high frequency which can be of the order of 0.1/0.2 eV.

-We've added the final paragraph to section 6, as follows: In an even higher energy range, we find a tunneling regime. For the electron to be injected into the environment beyond the CTS, its initial energy \(\varepsilon_I\) must first be reduced by exciting phonons on the CTS to get in resonance with the acceptor band. This situation is analogous to the crossing of a potential barrier, and the concept of Wigner time for barrier crossing can be applied here. It indicates that in this regime the electron spends more time on the CTS and can therefore be more sensitive to recombination. Additionally, as the density of states on the CTS decreases at high energies \(\varepsilon_I\), the injection time from the donor increases, according to Fermi's golden rule. Thus, in this tunneling regime, the residence times of the electron on the donor side and on the CTS are longer. Therefore if recombination processes are fast enough, they will reduce the quantum yield of the injection and possibly lead to a cold CTS. In the wide band recombination model the time for recombination is constant and will always become smaller than the Wigner time for residence on the CTS when \(\varepsilon_I\) increases. Yet in the narrow-band recombination model, the recombination time also increases with \(\varepsilon_I\). Therefore, once injected into the CTS, the electron may still have a significant injection yield on the acceptor side (see Figure \ref{figenvirV0dos} b))

-We've added the final paragraph to section 7, as follows: This study shows that the coupling between electrons and vibrational modes affects quantum efficiency, introducing the concepts of hot and cold charge transfer states. Our findings indicate that interface parameters are crucial for optimizing cell performance and that the interface's impact is more significant than environmental variations beyond the CTS. In addition, the electrostatic potential can play a moderate role in the injection process, even when it is attractive and creates localized bound electron-hole states at the interface. The charge injection process is described as a transfer into a continuum at an energy \(\varepsilon_I\). When \(\varepsilon_I\), is in the bottom part of the continuum an efficient resonant injection regime occurs with a cold CTS. At higher values of \(\varepsilon_I\) the electron must first loose energy by emitting phonons before going in the acceptor side. In this tunneling regime the electron spends more time on the donor orbital and then on the CTS site and the recombination can be favored.
Although a detailed comparison with a given system is beyond the scope of this work, we believe that the present model offers a simplified but comprehensive approach to advanced analyses of experimental results \cite{bassler2015hot}. It should be helpful for a better understanding of the exciton dissociation process and for finding ways to improve the organic solar cells. The present embedded charge transfer state model could still be improved by considering several orbitals and several modes per site and even finite temperature. We emphasize that the modeling of the dynamics of the recombination process is also an essential ingredient.

---

## Editorial Decision

published